# Drug Delivery Systems with a “Tumor-Triggered” Targeting or Intracellular Drug Release Property Based on DePEGylation

**DOI:** 10.3390/ma15155290

**Published:** 2022-07-31

**Authors:** Zhe Ren, Tao Liao, Cao Li, Ying Kuang

**Affiliations:** 1Ministry-of-Education Key Laboratory for the Green Preparation and Application of Functional Materials, Hubei Key Laboratory of Polymer Materials, Hubei University, Wuhan 430062, China; wangrenzheyi7010@163.com (Z.R.); ltvines@163.com (T.L.); 2National “111” Center for Cellular Regulation and Molecular Pharmaceutics, Key Laboratory of Fermentation Engineering (Ministry of Education), Hubei University of Technology, Wuhan 430068, China

**Keywords:** DePEGylationg, tumor-triggered targeting, dynamic protection strategy, controlled release

## Abstract

Coating nanosized anticancer drug delivery systems (DDSs) with poly(ethylene glycol) (PEG), the so-called PEGylation, has been proven an effective method to enhance hydrophilicity, aqueous dispersivity, and stability of DDSs. What is more, as PEG has the lowest level of protein absorption of any known polymer, PEGylation can reduce the clearance of DDSs by the mononuclear phagocyte system (MPS) and prolong their blood circulation time in vivo. However, the “stealthy” characteristic of PEG also diminishes the uptake of DDSs by cancer cells, which may reduce drug utilization. Therefore, dynamic protection strategies have been widely researched in the past years. Coating DDSs with PEG through dynamic covalent or noncovalent bonds that are stable in blood and normal tissues, but can be broken in the tumor microenvironment (TME), can achieve a DePEGylation-based “tumor-triggered” targeting or intracellular drug release, which can effectively improve the utilization of drugs and reduce their side effects. In this review, the stimuli and methods of “tumor-triggered” targeting or intracellular drug release, based on DePEGylation, are summarized. Additionally, the targeting and intracellular controlled release behaviors of the DDSs are briefly introduced.

## 1. Introduction

It was reported by Global Cancer Statistics 2020 that cancer has become the first or second leading cause of death before the age of 70 years in more than 110 countries in 2019, and there were an estimated 19.3 million new cases and 10 million cancer deaths worldwide in 2020 [1]. Due to disadvantages, such as poor solubility, unstable, unfavorable pharmacokinetics and poor biodistribution in vivo, lack of targeting capability, etc., traditional anticancer drugs show limited therapeutic effects with serious side effects [2]. Combined with nanotechnology, the anticancer drug delivery systems (DDSs) developed in recent decades have proved an efficient medium for targeted delivery and release of drugs to organelles, tissues, or cells, thus enhancing the stability, solubility, circulating half-life, and tumor accumulation of drugs [3,4,5]. As summarized by Shen et al., for efficient inhibition of a solid tumor, an excellent nanoparticle (NP)-based anticancer DDS should efficiently accomplish a process called the *CAPIR* cascade [6,7]. That is, the DDS should firstly have a long circulation (*C*) time in the blood compartments after being injected into the vein. After that, it needs to accumulate (*A*) in the tumor through the enhanced permeability and retention (EPR) effect; it then needs deep intratumoral penetration (*P*). Finally, the internalization (*I*) of the DDS by cancer cells is necessary for intracellular drug release (*R*) [6].

As the first step in the *CAPIR* cascade, enough circulation time of DDSs in vivo is important, it allows the DDSs to fully accumulate in tumor tissue through the EPR effect. However, after injection in the bloodstream, xenobiotics, such as DDSs, can interact with biomolecules, such as opsonins, which promote cellular opsonization and clearance by the mononuclear phagocyte system (MPS) [3,4,8,9]. Therefore, suitable protection strategies have to be introduced, in order to reduce the interaction between DDSs and MPS and make NPs “stealthy” [10]. For example, cell membranes are one kind of common “stealthy” coating, due to their functions, such as intercellular communication, bioantifouling, immune defense, etc. [10,11,12,13,14,15,16,17,18,19,20,21,22,23,24,25], including, but not limited to, membranes isolated from erythrocyte [15,16], monocyte [17], macrophage [18,19], neutrophil [20,21], lymphocyte [22,23], and platelet [24,25], which have been reported to mimic cells for DDS protection [10]. Similarly, the exosomes or extracellular vesicles derived from cells are also important “stealthy” coatings [26,27,28]. Besides biomembranes, some polymers with low protein absorption capability can also be used to prevent the conjugation of DDSs to proteins or cells, which protects DDSs from clearance [10]. As the electrostatic interaction between DDSs and organisms is important for circulation and cellular uptake, and negatively charged NPs have a longer circulation time, compared to positively/neutrally charged ones [29,30]; polyanions are considered to be a kind of effective “stealthy” coatings [30,31,32]. Compared to other “stealthy” materials, poly(ethylene glycol) (PEG), which has the lowest level of protein absorption of any known polymer, is undoubtedly the most widely used one [33,34,35,36]. PEG is a kind of highly flexible linear or branched synthetic polymer or oligomer with a molecular weight of 0.4-40 kDa; it is hydrophilic, with excellent biocompatibility, which is classified as generally regarded as safe (GRAS) by the Food and Drug Administration (FDA) [36,37]. It has been used in almost all kinds of DDSs, such as polymeric micelles [38,39,40], liposomes [41,42,43], hydrogels/nanogels [44,45,46], polymeric NPs [47,48], inorganic NPs [49,50], aggregation-induced emission luminogens (AIEgens) [51], etc. As a kind of hydrophilic polymer, PEG can also improve the water solubility, dispersibility, and stability of DDSs both in vitro and in vivo [37]. Since it was firstly reported to modify proteins (so-called PEGylation) in 1977, tens of thousands of research works on PEG-based DDSs have been published, and tens of PEG-based nanomedicines have been approved by the FDA [5,35].

However, though “stealthy” coatings represented by PEG can prolong the circulation time of DDSs in vivo and improve their pharmacokinetics, “stealthy” coatings can also diminish cellular uptake [29,31,52], which interferes with step *I* in the *CAPIR* cascade and may reduce the therapeutic effect of DDSs. As a result, dynamic protection strategies have been developed in the last decade. The charge reversal strategy, based on tumor-acidity-cleavable maleic acid amide (TACMAA), is a representative one [31]. For amino group-based polycations coated on the surface of DDSs, modification of the amino groups with 2,3-dimethylmaleic anhydride or citraconic anhydride can change them to carboxyl-contained polyanions. The hydrolysis of the TACMAA in the slightly acidic tumor microenvironment (TME) can return the acid amide to the amino group, thus achieving a negative-to-positive charge-reversal, which activates the “stealthy” DDSs to “visible” to the cancer cells [31,53,54,55,56]. Another common dynamic protection strategy is DePEGylation, that is, stripping the “stealthy” PEG coating after DDSs accumulation in tumors, with the help of TME (Figure 1) [57]. Based on the DePEGylation strategy, Zhang’s group proposed a concept called “tumor-triggered” targeting, or intracellular drug release, in 2010 [58]. It offers a new way to design DDSs, with both long circulation time in vivo and good internalization capability by cancer cells using the most popular “stealthy” coating PEG. In this review, we try to briefly introduce the development of DDSs with “tumor-triggered” targeting or intracellular drugs or genes, a kind of macromolecular “drug”, release property based on DePEGylation in the last decade. Additionally, the targeting and intracellular controlled release behaviors are briefly introduced.

## 2. pH-Sensitive DePEGylation for “Tumor-Triggered” Targeting or Intracellular Drug Release

Due to the non-fouling feature of the PEG molecular chain, PEGylation mainly depends on PEG’s functional end group(s). To modify DDSs with PEG, there are three typical methods, as follows [59]. (1) Adsorption of PEG through noncovalent bonds, such as the hydrogen bond and electrostatic and coordination interaction, on the surface of DDSs. (2) Grafted PEG on the surface of DDSs through covalent bonds. (3) Linking PEG as the hydrophilic block(s) to other polymeric chains to form block copolymers, for instance, amphiphilic polymers that can self-assemble into PEGylated NPs (micelles, vesicle, etc.) in water [60]. For PEGylation based on noncovalent adsorption, the differences (in most cases, pH) between the environment in blood/normal tissues and TME may destroy or weaken the noncovalent interaction, thus leading to DePEGylation [61]. For the other two PEGylation strategies, PEG could be grafted on the surface of DDSs or linked with other polymeric chains through dynamic chemical bonds, which are reversible in blood/normal tissues and TME. Endogenous stimuli, such as pH, enzyme, and redox, and exogenous stimuli, such as light, can drive the broken of the dynamic bonds in tumor tissues, resulting in DePEGylation.

Compared to normal cells, cancer cells usually ingest and consume more glucose, and the aerobic glycolysis, or so-called Warburg effect, in cancer cells produces a large amount of lactate, which was discharged into the extracellular matrix (ECM) and reduced tumoral extracellular pH (pH_e_) [62]. As a result, pH-sensitive noncovalent and covalent bonds are widely introduced for DePEGylation.

### 2.1. Benzoic Imine Bond-Based DePEGylation for “Tumor-Triggered” Targeting or Intracellular Drug Release

In 2008, Gu et al. reported a micellar DDS self-assembled from an amphiphilic polycation, cholic acid (CA), and PEG-grafted poly-_L_-lysine (PLL), with PLL-g-CA as a hydrophobic block, and PLL and PLL-g-PEG as hydrophilic blocks. PEG was grafted to PLL via a pH-sensitive benzoic imine bond formed from benzaldehyde and amino group (Figure 1). The benzoic imine bond is stable in neutral and alkaline environments but can be hydrolyzed in a weakly acidic environment (pH < 6.8); as a result, the micelles were “stealthy” at physiological pH but could reveal the positive charge in a weakly acidic environment, e.g., TME, for better uptake by cancer cells [63,64,65]. Based on this work, Zhang’s group proposed the concept of “tumor-triggered targeting” in 2010 [58]. In this work, they reported the noncovalently connected micelles (NCCMs) for anticancer drug delivery. As shown in Figure 2, hydrophilic poly(N-isopropylacrylamide-co-N-acroyloxysuccinimide) (P(NIPAAm-co-NAS)) and hydrophobic poly(ε-caprolactone) (PCL) were linked through an α-β cyclodextrin (CD) dimer via host–guest interaction to form an amphiphilic copolymer that could self-assemble to micelles. A cancer cell over-expressed α_v_β_3_ integrins targeting peptide, glycine-arginine-glycine-aspartic acid-serine (GRGDS) [66], was coated on the hydrophilic shell of the micelles and shielded by methoxy PEG benzaldehyde (mPEG-CHO) through a benzoic imine bond. DePEGylation, caused by the weakly acidic environment, led to the revealing of GRGDS and achieved a “tumor-triggered targeting”. PNIPAAm is temperature-sensitive with a lower critical solution temperature (LCST)~32 °C and could be adjusted to 38 °C by copolymerization with hydrophilic NAS [67]. Therefore, the NCCMs could shrink and disassemble in the overheating cancer cells to release the loaded drug doxorubicin (DOX).

After that, the method of “tumor-triggered targeting” has been widely applied to the design and fabrication of novel anticancer DDSs. For example, Yang et al. reported a kind of mixed micelles self-assembled from two amphiphilic polymers cyclic RGD (cRGD) functionalized PEG-b-PCL (cRGD-PEG-b-PCL) and benzoic imine-linked PEGylated DOX (PEG-DOX). The micelles were “stealthy” by the protection of PEG-DOX and could turn to cancer cell-affinitive due to the DePEGylation of PEG-DOX in TME to reveal the targeting cyclic peptide [68]. The loaded anticancer drugs DOX and paclitaxel (PTX) could be released intracellularly, due to the acid-trigged structure changes.

In 2014, Zhang’s group further introduced this method to mesoporous silica nanoparticle (MSN)-based DDS [69]. MSN is a kind of hot material, due to its good biocompatibility, plasticity of size, shape, and pore size, easy surface modification, high drug loading capability, etc. [70,71], and it is well-studied in the drug delivery field. In this work, they loaded MCM-41-typed MSN with DOX, then used peptide RGDFFFFC as the “gatekeeper”, and shielded it by mPEG-CHO through a benzoic imine bond. The peptide contained the targeting sequence RGD, the four-phenylalanine (FFFF) sequence to block the pores through the π-π stacking and hydrophobic interactions between the benzene rings, as well as a cysteine (C) residue to linked with MSN through a redox-sensitive disulfide bond. DePEGylation resulted in “tumor-triggered targeting”. The addition of disulfide reducing agents DTT (to imitate reduced glutathione (GSH) intracellular) could break the disulfide bond, thus leading to DOX release. Our group also reported the “tumor-triggered targeting” MSN-based DDSs with chitosan (CHI) as the “gatekeeper” [72,73]. CHI is the only natural alkaline polysaccharide with low toxicity, excellent biodegradability, and low immunogenicity; it is obtained from organisms such as the shell and cartilage of arthropods and mollusks, and the cationic polysaccharide characteristic allows CHI to stick to the cell membrane and enter cells via electrostatic interaction [74,75]. As shown in Figure 3, we prepared β-CD-grafted CHI (CHI-g-CD) and coated it on DOX-loaded MSN through host–guest interaction between β-CD and benzimidazole (Bz). After that, CHI was grafted with peptide GRGDS and then mPEG-CHO through a benzoic imine bond, respectively. Besides the “stealthy” characteristic, PEG could also enhance the stability of the NPs (the particle size kept ~260 nm within a week in the culture medium containing 10% fetal bovine serum (FBS)). After the DePEGylation, both GRGDS and the positive charge of CHI promoted the uptake of the NPs. Then, the more acidic environment intracellular destroyed the host–guest interaction between β-CD and Bz [76,77], thus leading to the departure of CHI for drug release. The targeting peptide also ensured the better endocytosis capability of the NPs by mouse breast cancer (4T1) cells than that of normal cells of human renal epithelial (293T). Both in vitro and in vivo experiments proved that DDSs could improve the utilization of DOX and reduce its side effects.

Zhang’s group also introduced “tumor-triggered targeting” into metal-organic-framework (MOF)-based DDS in 2015. In this work, they applied a one-pot and organic solvent-free “green” post-synthetic method to fabricate a MOF/β-CD/peptide-PEG DDS [78]. The nanosized iron-based MOF MIL 101 with a mesoporous structure was used to load DOX. β-CD acted as the “gatekeeper” to block the pores through a disulfide bond to avoid premature drug release. Peptide-PEG copolymer K(ad)RGDS-PEG1900, which was a cancer-targeting peptide KRGDS (K stands for lysine) with adamantine (Ad) modified on the K terminal, and PEG modified on the S terminal was introduced as the “tumor-triggered targeting” dynamic protection layer. K(ad)RGDS-PEG1900 was coated on the NPs through the host–guest interaction between β-CD and Ad, and PEG was linked with the peptide through a benzoic imine bond. The DePEGylation in the weakly acidic TME revealed the peptide KRGDS for cancer targeting, and the interruption of the disulfide bond intracellular led to the leave of β-CD to “open the gate” for DOX release.

Besides”tumor-triggered targeting”, “tumor-triggered” intracellular drug release based on benzoic imine bond has also been reported. By utilizing the supermolecular (host–guest) interaction between mono-6-deoxy-6-EDA-β-CD (β-CD-NH_2_) and the guest molecules, Wang et al. prepared a series of DOX-loaded supramolecular NPs [79,80]. β-CD-NH_2_ also offered an amino group for the grafting of mPEG-CHO through the benzoic imine bond. DePEGylation of the NPs in the weakly acidic TME revealed the positively charged amino groups to promote the uptake of the NPs by cancer cells via electrostatic interaction. Then, the more acidic environment intracellular broke the linkage of the guest molecules (interrupting pH-sensitive acetal or boronate ester bonds), thus leading to the dissolution of the NPs and DOX release. Our group also applied β-CD-NH_2_ as the “gatekeeper” to block the pores via acetal or boronate ester bonds to prepare the dual-pH-sensitive MSN-based DDSs with mPEG-CHO coated through the benzoic imine bond, which could also achieve a “tumor-triggered” intracellular drug release [81,82]. Recently, Hung et al. reported a polymeric vehicle-based DDS (ID@PbCTPNs) fabricated from the co-assembly of amphiphilic PEG-benzoic imine-octadecane (mPEG_5k_-b-C_18_) and tocopheryl PEG succinate (TPGS), as well as hydrophobic poly(lactic-co-glycolic acid) (PLGA) for both photothermal agent indocyanine green (ICG) and drug DOX delivery [83]. With the pH decreased from 7.4 to 6.5 and then 5.0, ID@PbCTPNs could drop their PEG protection layer for cellular uptake and reduce the particle size for both drugs release intracellular in succession, thus achieving a photothermal/chemo-combinatorial cancer therapy.

### 2.2. TACMAA-Based DePEGylation for “Tumor-Triggered” Targeting or Intracellular Drug Release

The negative-to-positive charge reversal strategy, based on TACMAA, is an important dynamic protection strategy [31]. It is expected that both the pH-trigged charge reversal and the break of TACMAA could be introduced to DePEGylation. In 2014, Zhang’s group reported a TACMAA charge reversal-based “tumor-triggered” intracellular drug release DDS, named MSN-TPep/PEG-PLL(DMA), for the co-delivery of drug and therapeutic peptide to mitochondria with a DePEGylation feature in TME. As shown in Figure 4, they firstly loaded MSN with anticancer drug topotecan (TPT), then used triphenylphosphonium (TPP)-modified antibiotic peptide (Lys-Leu-Ala-Lys-Leu-Ala-Lys)_2_ (Tpep) as the “gatekeeper” to block the pores via a disulfide bond; after that, they applied PEG-b-2,3-dimethylmaleic anhydride-modified PLL (PEG-PLL(DMA)) to shield the positive charge of the nanoparticles through the electrostatic interaction between TPP and PLL(DMA) [84]. The charge reversal of PLL(DMA) in TME resulted in the electrostatic repulsion of the copolymer and the nanoparticles, thus leading to DePEGylation. After entering the cancer cells with the help of the positive charge, the DDS could release both TPT and Tpep, due to the redox sensitivity of the disulfide bond. The mitochondria-targeted TPP brought the Tpep to specific mitochondria-induced damage, and TPT could break the nucleus. The dual-sensitive and combination therapy characteristics made the system a nice choice for safe and effective cancer therapy. Similarly, Zhao et al. grafted a β-carboxylic amide functionalized PEG (PEG-DA) layer on the poly(ethylenimine) (PEI)-poly(trimethylene carbonate)/DNA (PEI-PTMC/DNA) complex nanoparticles through the electrostatic interaction, which achieved a “tumor-triggered” intracellular DNA release for cancer gene therapy [85].

Wang’s group who first introduced the TACMAA-based charge reversal strategy also reported the TACMAA-based chemical DePEGylation [86,87]. They prepared the nanoparticles based on a kind of block copolymer with PEG linking to poly(d,l-lactide) (PDLLA) through the TACMAA, which were used for docetaxel (DTXL) delivery. Similar to other pH-sensitive chemical bonds, such as benzoic imine, the break of TACMAA in acidic TEM could result in DePEGylation and a rise of the zeta potential, which significantly enhanced the cellular uptake and in vivo cancer therapy effect, compared to the control system without pH-sensitivity. Then, Zhou et al. grafted PLL-lipoic acid (PLL-LA) with PEG through TACMAA to prepare a nanosystem for DOX and programmed death receptor 1 (PD-1) small interfering RNA (siPD-L1) delivery, thus achieving a “tumor-triggered” intracellular both drug and gene release for combined chemotherapy and gene-mediated immunotherapy [88]. In 2020, this strategy has also been introduced to a tumor photothermal therapy (PTT) system. By coating of CHI-wrapped carbon nanotubes (CHI-CNTs) with TACMAA-modified eight-armed PEG, a “tumor-triggered targeting” photothermal delivery nanoplatform was obtained. The DePEGylation resulted in a highly effective uptake of the nanoparticles by murine bladder cancer cells (MB49) and mitochondrial targeting [89]. Then, the photothermal effect of CHI-CNTs under irradiation with a near-infrared (NIR) light (808 nm) could damage the mitochondria, which further induced the reactive oxygen species (ROS) for cell death. Wang et al. applied photosensitizer zinc phthalocyanine (ZnPc) as a drug and bridge, thus linking PEG through the TACMAA and hyaluronic acid (HA) chain through a disulfide bond to obtain the copolymer PEG_4000_-CDM-ZnPc-S-S-HA, which could assemble into nanoparticles for tumor passive targeting [90]. The DePEGylation revealed HA, a cancer cell overexpressed CD44 receptor targeting polysaccharide [91], thus achieving a “tumor-triggered targeting”. ZnPc could be released intracellularly, due to the high concentration of GSH for NIR light-triggered photodynamic therapy (PTT). By mixing PEI-crosslinked β-CD (PEI-β-CD) and TACMAA-linked adamantyl PEG (Ad-CDM-PEG) through the host–guest interaction, Tang’s group prepared a polymer-based nanoplatform PCACP for microRNAs, miR-199a/b-3p mimics (miR199), and antimiR-10b (antimiR10b) delivery for hepatocellular carcinoma (HCC, one of the most malignant primary cancers) cocktail therapy [92]. The introduction of Ad/β-CD host–guest interaction made the surface modification of dynamic PEG chains easier, and the TACMAA-based DePEGylation strategy, as well as the cocktail therapy, could effectively inhibit tumor cell proliferation, migration, and invasion, as well as reduce the side effects of normal cells.

### 2.3. Ortho Ester-Based DePEGylation for “Tumor-Triggered” Targeting or Intracellular Drug Release

Ortho ester is another kind of pH-sensitive bond that can be broken in a slightly acidic environment and used for DePEGylation and controlled drug release [93]. Based on ortho ester, amphiphilic graft copolymer dynamic micelles with polyurethanes or poly(disulfide)s backbones and PEG hydrophilic side chains were reported [94,95]. The ortho ester-linked PEG could be stripped at pH 6.5, resulting in the size changes of the micelles, which allowed the micelles to penetrate the centers of SH-SY5Y multicellular spheroids quickly for controlled intracellular drug release and enhanced chemotherapy. The same group also reported a kind of anticancer PEG-modified nanogels through ortho ester with a “tumor-triggered targeting” property [96,97]. As shown in Figure 5, they used four functional monomers, i.e., 4-(hydroxymethyl)-phenylboronic acid pinacol ester, 3-(acrylamido)phenylboronic acid slightly acid-sensitive, PEG derivative, and cisplatin (Pt(IV)) crosslinker, for copolymerization to obtain the dynamic PEG-modified nanogels. After accumulation in tumor tissues through the EPR effect, the breaking of the ortho ester bond led to DePEGylation to reveal phenylboronic acid (PBA) for cancer cells overexpressed in sialic acid active targeting. The nanogels could further deplete the GSH intracellular by the dual-regulation of Pt(IV) and boronic acid pinacol ester, which could enhance the therapeutic effect of the released Pt(II) reduced from Pt(IV). Additionally, this kind of nanogel-based DDS showed a high accumulation in the tumor region (6.4% ID/g), due to both passive and “tumor-trigged” active targeting. In 2021, they also copolymerized diamine monomer of ortho ester with drug derivatives (modified Pt(IV) or methotrexate (MTX)) and then terminated the copolymers by PEG-active ester to obtain dynamic drug-backboned nano-prodrugs, which could self-assemble into micelles for passive targeting and “tumor-trigged” intracellular drug release [98,99].

### 2.4. Other pH-Sensitive Linkages-Based DePEGylation for “Tumor-Triggered” Targeting or Intracellular Drug Release

Similar to benzoic imine, both imine (-N=CH-) and hydrazide-hydrazone (-CO-NH-N=CH-) are pH-sensitive chemical bonds responsive to the pH changes in tumor tissue [100] and can also be used for DePEGylation. For example, a kind of polysaccharide-based polymeric lipid vesicles was designed for “tumor-triggered targeting” in 2014 [101]. Folate acid (FA)-modified dextran was grafted with hydrophobic stearyl alcohol through a disulfide bond and PEG through a hydrazone bond, and the amphiphilic macromolecules could self-assemble into vesicles to load DOX. After enrichment in tumor tissues through EPR effect-trigged passive targeting, the hydrolysis of the hydrazide-hydrazone bond led to DePEGylation to reveal the active targeting ligand FA, thus achieving a “tumor-triggered targeting”. Additionally, the vesicles could further release loaded DOX intracellular, due to the redox sensitivity. Similarly, Sun et al. synthesized a series of FA-modified amphiphilic block-statistical copolymers, monomethoxy PEG-FA-poly(oligo(ethylene glycol)monomethyl ether methacrylate)-st-poly(2-hydroxyethyl methacrylate-g-lactide) (mPEG(FA)-P(OEGMA300)-st-P(HEMA-g-LA)) that could self-assemble into micelles, with PEG shielding FA through hydrazone [102]. After acidic “tumor-triggered targeting”, the redox-sensitive micelles could intracellularly release drugs. This method could also be introduced to fabricate liposome-based DDSs [103]. Cheng et al. reported the FA and PEGylated phospholipid-modified ammonium bicarbonate (ABC) liposomes, in which PEG was linked with phospholipid through an imine bond that made the traditional ABS liposomes, which could only release drugs extracellularly and bring DOX into the cancer cells, due to the “tumor-triggered targeting” property [104]. Additionally, the acid (pH 6.4), as well as the high temperature (42 °C), led to the production of CO_2_ bubbles, which trigged the DOX release intracellular. Recently, Domiński et al. also prepared copolymer micelles with the hydrophilic block PEG linking to hydrophobic block through a hydrazone bond, which could be hydrolyzed under an acidic pH for DePEGylation and achieve an intracellular drug release [105].

Another kind of pH-sensitive bond, phenyl-substituted vinyl ether (PIVE), was introduced by Kim et al. for DePEGylation. In these works, PEG has been linked with 1,3-dioctadecyl-rac-glycerol (DOG) lipids through PIVE, and the obtained PEG-PIVE-DOG was implanted in the liposome-based drug or gene delivery system. Hydrolysis of PIVE bond in the acidic environment led to DePEGyltion and, more importantly, the pH-sensitivity of PIVE could be adjusted by rational selection of the phenyl ring substituent [106,107].

Protonation of acid radicals in an acidic environment can reduce the electrostatic interaction between the acid radicals and cations. As a result, Luo et al. synthesized 12-hydroxystearic acid-PEG-YGRKKRRQRRR (HA-PEG-TAT) to load disulfiram (DSF), in the form of cationic nanoparticles, and then coated them with negatively charged poly(glutamate acid)-graft-PEG (PGlu-PEG) through the electrostatic interaction between glutamic acids and arginine [108]. At pH 6.5, the protonation of glutamate acid radical weakened the electrostatic interaction; PGlu-PEG transformed into α-helix and left the nanoparticles to achieve a charge reversal. Then, the cell-penetrating peptide TAT induced the membrane tension for DSF releasing intracellular.

Coordination is another interaction for PEG grafting. Based on coordination, Tseng et al. prepared a PEI-based polymeric gene delivery system. They synthesized the histidine-grafted PEI (hisPEI) and nitrilotriacetic acid end-capped PEG, followed by chelation with Zn^2+^. After the formation of hisPEI/DNA complexes, PEG was coated through the coordination interaction between Zn^2+^ and histidine. At pH 4.0, the breaking of the coordination interaction led to DePEGylation for intracellular DNA release and gene therapy [109].

## 3. Other Stimuli-Sensitive DePEGylation for “Tumor-Triggered” Targeting or Intracellular Drug Release

### 3.1. Enzyme-Sensitive “Tumor-Triggered” Targeting or Intracellular Drug Release

Another difference between tumors and normal tissues is the overexpression of enzymes, such as matrix metalloproteinases (MMPs), caspases, phospholipases, and hyaluronidases [110]. MMPs are a kind of zinc-dependent protease enzyme that can degrade the ECM components; among MMPs, both MMP-2 and MMP-9 are overexpressed in many kinds of cancers and often explored for enzyme-triggered, controlled drug release [110,111]. Linking the “stealthy” polymers on the surface of the DDSs through MMP degraded peptides, such as Pro-Leu-Gly-Val-Arg (PLGVR), can achieve MMP-triggered DePEGylaytion or dynamic protection [29,112].

Based on this method, Zhang’s group reported a PEG/β-CD/MSN complex DDS with the “gatekeeper” β-CD, coated on MSN, to block the pores through a thermal-labile Azobenzene (Azo) linker and GGPLGVRGRGDK-Ad peptide-linked PEG-grafted outside via host–guest interaction. The cleavage of PLGVR in the existence of overexpressed MMP-2 led to DePEGylation and the revealing of the targeting peptide RGD, thus resulting in a “tumor-triggered targeting” [113]. Photothermal agent ICG and drug DOX were loaded in this DDS; as a result, NIR exposure aroused the photothermal effect of ICG and broke the Azo linker to “open the gate” for DOX release, thus achieving a combined chemo/photothermal-therapy. Similarly, Ge’s group reported a series of amphiphilic block copolymers, which could self-assemble into micelles or vesicles to load drugs, in which hydrophilic PEG linked with hydrophobic segments via MMP-2-sensitive peptide GPLGVRGDG; these systems could also achieve the “tumor-triggered targeting” and intracellular DOX release [114,115,116]. These kinds of block copolymers could also be used for intracellular gene therapy, with a “tumor-triggered targeting” property [117]. As shown in Figure 6, Gordon et al. prepared a kind of polymethacrylate-based nanogels, with polymer chains crosslinked by disulfide bonds and surface modification by GPLG-linked PEG. After passive targeting of tumor tissues through the EPR effect, the cleavage of GPLG by MMP-9 could reveal the positive charge on the surface of the nanogels for better cellular uptake, as well as the breaking of disulfide bonds in a high concentration of GSH disintegrating the nanogels and resulting in drug release intracellular [118].

To prevent the metastatic and local-regional recurrence of cancer after surgery, Li et al. reported a polymersome-based DDS, assembled from a triblock copolymer PEG-GPLGVRG-PCL-PGPMA, composed of MMP-cleavable peptide-linked PEG and PCL, and CPP-mimicking polymer poly(3-guanidinopropyl methacrylamide) for codelivery of microtubule-disrupting and anti-inflammatory agent colchicine and MMP inhibitor marimastat [119]. In TME, the overexpressed MMPs led to the DePEGylation for not only the transformation of the polymersomes into multicompartmentalized nanoassemblies (revealing guanidine groups for tissue/cell-targeting), but also colchicine and marimastat release. After the surgical removal of large primary tumors, the released drugs from the DDS could influence the generation of microtubules and inhibit overexpressed MMPs, thus reducing metastatic and local-regional recurrence [119].

In 2015, Bruun et al. reported a titratable cationic lipid nanoparticle-based SiRNA delivery system that could cross the impermeable blood-brain barrier (BBB) with a DePEGylation property [120]. By coating the lipid nanoparticles with both angiopep-terminated 1,2-distearoyl-sn-glycero-3-phosphoethanolamine-PEG_2000_ (DSPE-PEG_2000_) and dimyristoyl-PEGylated cleavable lipopeptide (DM-PCL), the carriers could anchor the overexpressed low-density lipoprotein receptor-related protein-1 (LRP-1) through the targeting ligand angiopep firstly and then take off the PEG protection layer, due to the MMP-degradation of the GWIPVSLRSGEE peptide of DM-PCL for uptake by brain endothelial and glioblastoma cells, which achieved the crossing of the BBB. The MMP-triggered dePEGylation substantially increased the uptake and gene silencing efficiency of the carriers.

### 3.2. Redox-Sensitive “Tumor-Triggered” Targeting or Intracellular Drug Release

Glutamate–cysteine–glycine, or GSH, is an important reduced tripeptide and the most abundant intracellular small molecule biothiol, which plays an important role in some cellular processes, such as maintaining the redox balance intracellular [121]. A feature of GSH is that the GSH concentration intracellular (1–0 mM) is considerably higher than that of extracellular (1–10 µM), and GSH is overexpressed inside certain cancer cells, which makes GSH an ideal stimulus for designing redox-sensitive DDSs with an intracellular controlled release behavior [121,122]. However, different from other strategies, GSH-based, redox-sensitive DePEGylation should happen inside the cancer cells, thus resulting in “tumor-triggered” intracellular drug release.

In 2010, Li’s group reported an “intelligent” nanoassembly (INA) for gene delivery. It contained a plasmid DNA (pDNA)/protamine sulfate (PS) condensed core and dioleoylphosphatidyl ethanolamine (DOPE)-based lipid envelope with PEG linked on the DOPE through a GSH-sensitive disulfide bond [123]. PEG could stabilize the structure of INA and prolong its systemic circulation time. Additionally, the DePEGylation of the INA in the reductive intracellular environment led to the broken structure of INA and a tumor-triggered release of pDNA. Additionally, this system showed a better transfection efficiency and lower cytotoxicity than that of the commercialized gene delivery reagent Lipofectamine^TM^ 2000. Then, Zhang’s group reported a corona-shell-core copolymer PEG-b-poly(L-lysine)-b-poly(racleucine) (PEG-SS-PLys-PLeu), with PEG linked with the poly(amino acid) chains through a disulfide bond. As shown in Figure 7, the copolymer could self-assemble into micelles to load DOX, and the PLys shell of the micelles was also cross-linked by disulfide bonds. In the presence of a high concentration of GSH intracellular, PEG could leave the micelles, and the micelles could collapse to release DOX [124]. Similarly, Ai et al. prepared star-shaped, disulfide bond-linked PEG-Lysine-di-tocopherol succinate copolymer-based micelles for “tumor-triggered” intracellular DOX release [125]. Besides the TACMAA-based nanoplatform PCACP fabricated by Ad-CDM-PEG and PEI-β-CD [92], Tang’s group also reported a GSH-sensitive DePEGylation Ad-SS-PEG/PEI-β-CD-based gene delivery system for HCC therapy [126].

Similar to disulfide, the diselenide bond is GSH-sensitive and can be cleaved by GSH intracellular [122,127,128]. Li et al. also prepared a diselenide-linked polycation mPEG-SeSe-PEI for “tumor-triggered” intracellular DNA release [129].

Hypoxia is a noteworthy characteristic observed in most solid tumors, due to the demand for a large number of nutrients and oxygen by the rapidly proliferating cancer cells, as well as the structurally defective and irregular tumor micro-vessels [130]. Azo is a hypoxia-sensitive group and has been used for DePEGylation, too. In 2018, Li et al. synthesized a copolymer methoxy PEG-Azo-poly(aspartic acid), with imidazole as the side chains, that could self-assemble into conjugate micelles to load photosensitizer chlorin e6 (Ce6) [131]. In the anoxic tumor tissues, the collapsing of Azo led to DePEGylation to facilitate cellular uptake. Irradiation of Ce6 by NIR (660 nm) produced singlet oxygen, which oxidated imidazole to disassemble the micelles for Ce6 release. As a result, the intracellular Ce6 concentration increased, oxygen level was decreased, and diffusion range and half-life of singlet oxygen were enhanced. Additionally, this system showed a good anticancer photodynamic therapy (PDT) effect. Im et al. loaded Ce6 in MSN, which was coated with glycol CHI (GC) and PEG through Azo linker, then adsorbed CpG oligonucleotide through the electrostatic interaction with GC to obtain a DDS, named CAGE, for PDT-enhanced cancer immunotherapy (CIT) [132]. The leaving of both GC and PEG facilitated the uptake of CAGE by dendritic cells (DCs), and the released CpG could bind toll-like receptor 9 (TLR 9) and activate the DCs, as well as upregulate costimulatory markers essential for antigen presentation. Additionally, the reactive oxygen species (ROS) obtained by irradiation of Ce6 could generate immunogenic debris, recruit DCs, and kill cancer cells.

### 3.3. Light-Trigged DePEGylation for Targeting or Intracellular Drug Release

Light, especially NIR light, is a kind of important exogenous stimuli in anticancer fields. Photosensitizer-based PDT, photothermal material-based photothermal therapy (PTT), and light-trigged controlled release are widely studied, as light-based cancer therapy is non-invasive, non-destructive, and non-ionizing [133,134,135]. Though light is not a biosignal in TME, it can be controlled to target irradiated tumor tissues; therefore, the light-trigged DePEGylation for targeting or intracellular drug release is also briefly introduced in this review. In 2012, Ji’s group reported a simple light-trigged DePEGylation DDS for intracellular DNA release [136]. In this work, they synthesized the β-CD-modified PEI (PEI-CD) and Azo-modified PEG (Az-PEG), grafted the Az-PEG to PEI-CD via host–guest interaction between β-CD and Azo, and then packed DNA through electrostatic interaction. Besides being hypoxia-sensitive, Azo is also light-sensitive and can be reversibly isomerized from *trans* isomer to *cis* isomer by irradiation at 365 nm, and it loses its host–guest interaction with β-CD [136,137]. As a result, the ultraviolet (UV) light could trigger the DePEGylation of the gene carriers for the DNA release intracellular.

Other light-trigged DePEGylation strategies are mainly based on light-cleaved covalent bonds. For example, Li et al. synthesized two amphiphilic copolymers, i.e., the nucleus localization sequence (NLS) PKKKRKVR peptide-linked 1,2-distearoyl-sn-glycero-3-phosphoethanolamine-N-(PEG2K) (DSPE-PEG2K-NLS) with a short PEG chain (molecular weight 2k Da) and photoresponsive prodrug PEG5K-[7-(diethylamino)coumarin-4-yl]methyl (DEACM)-Chlorambucil (Cb) (PEG5K-DEACM-Cb) with a long PEG chain (molecular weight 5k Da), which were assembled into the long-chain PEG5K shielded nanoparticles [138]. The irradiation of 420 nm LED light triggered the breakdown of the linkage between DEACM and Cb, thus resulting in DePEGylation of PEG5K. After entering the cancer cells, targeted NLS brought the nanoparticles into the nucleus to release Cb, an anticancer drug that kills cancer cells by interfering with DNA.

As shown in Figure 8, Zhou et al. prepared a hydrophilic PEG and hydrophobic pH-sensitive poly(β-aminoester)s (PAE) containing amphiphilic polymer PEG-Nbz-PAE-Nbz-PEG with the o-nitrobenzyl (Nbz) linkers connected the blocks and cRGD-modified PAE, which self-assembled into micelles to load both DOX and NaYF_4_:Yb/Tm@NaYF_4_ up-conversion nanoparticles (UCNPs). The UCNPs could convert the irradiated NIR to UV light to break the Nbz linkers for DePEGylation of the micelles [139], thus achieving a reversed cRGD-mediated cancer targeting, better vascular extravasation, and deep tumor penetration [140]. The pH-sensitive of PAE led to the disassembly of the micelles and intracellular DOX release. By using of UCNPs, the cleavage of Nbz was transformed from UV light to NIR and more suitable for in vivo applications. By application of Nbz, a light-trigged targeting liposomal DDS was also reported, and the drug delivery behavior was studied in a zebrafish xenograft model [141].

Zhu et al. applied an indirect method to achieve a light-triggered DePEGylaiton in 2019 [142]. They also synthesized the amphiphilic light-sensitive thioketal (TK) bond-linked PEG and Ce6 (mPEG-TK-Ce6), which was used to assemble into nanoparticles with polylactide (PLA) to load the Pt(IV) prodrug. After the accumulation of the nanoparticles in tumor tissues, the irradiation at 660 nm stimulated the generation of ROS by Ce6, which could break the TK link for DePEGylation and an enhanced cellular uptake, and then the redox-sensitive Pt(IV) released intracellular to kill the cancer cells.

## 4. Conclusions and Perspectives

As one of the most important strategies to protect nanosized anticancer DDSs, PEGylation has been widely used for more than 40 years. From primary liposomes and micelles to multifunctional compounded nano-systems, their solubility and dispersibility in water can be effectively enhanced by PEG. What is more, the characteristic of the lowest level of protein absorption of any known polymer makes PEG an excellent “stealthy” coating to help DDSs to escape from the clearance by the MPS, which can improve their pharmacokinetics, prolong their blood circulation time and enhance their accumulation in tumor tissues through the EPR effects. Though PEG is a kind of synthetic polymer, compared to other common materials used in drug delivery fields such as biopolymers [143,144,145], its biosafety is also approved by the FDA. Additionally, the better “stealthy” character makes PEG better protective coatings than biopolymers [143]. However, the shortcomings of PEGylation are also nonnegligible. Compared to the abundant functions such as cancer targeting and biodegradation of biopolymers represented by polysaccharides, proteins, and nucleic acid [143,144,145], no more functions, besides “stealth”, can be offered to DDSs by PEG. On the other hand, though the cost of synthesizing unfunctionalized PEG is not high, the functionalization of its terminal group(s) is costly. In contrast, biopolymers are rich in sources and easy to be extracted from plants and animals, rich in hydroxyl, carboxyl, aminos, and other groups, thus making them easily chemically-modified; many of them have been industrialized and are low cost. Most importantly, the lacking active targeting makes PEGylated DDSs hard for uptake by cancer cells, which reduces the utilization of drugs.

To keep the advantage of the “stealth” of PEG and overcome the disadvantage of hindering endocytosis by cancer cells, dynamic protection strategies were introduced to DDSs. Among these strategies, the DePEGylation-based “tumor-triggered” targeting or intracellular drug release has been developed for more than 10 years and one of the important methods to dynamically protect the nanosized anticancer DDSs, as well as endow the DDSs with the capability of full circulation in the blood compartments to accumulate in tumor tissues through the EPR effect and enhanced internalization by cancer cells through active targeting or electrostatic interaction for intracellular drug or gene release, which are usually considered contradictory. As summarized in Table 1, nowadays, endogenous stimuli, such as pH, enzyme, redox, and exogenous stimuli, such as light, have been applied for DePEGylation by cleaving the dynamic chemical bonds, such as benzoic imine, TACMAA, ortho ester, MMP-sensitive peptide, Azo, etc., or destroying the noncovalent interaction, such as the electrostatic or host–guest interactions. Additionally, revealing targeting ligands, such as RGD peptide, HA, and FA, etc., or positive charge can significantly improve the uptake capability of cancer cells. As it is difficult for traditional nanosized anticancer DDSs to obtain both “stealthy” property for long blood circulation and good cancer cells endocytosis capability, DePEGylation-based “tumor-triggered” targeting or intracellular drug release offers a good idea to further enhance the utilization of anticancer drugs, as well as reduce their side effects, and shows excellent prospects in both scientific research and clinical application fields.

Of course, DePEGylation-based “tumor-triggered” targeting or intracellular drug release strategies still have some shortcomings in the fabrication of dynamic DDSs. One of them is the further enlarged synthetic difficulties. As certain chemical structures are needed for DePEGylation, the synthesis of both specific functional PEG and PEG-coated nanoparticles is more difficult than that of non-dynamic PEGylation DDSs. This also increases the synthesis costs and makes clinical application difficult. Therefore, it is expected that other bonds or interactions could be introduced in this field to simplify the synthesis process and reduce the cost, in order to meet the needs of clinical use. On the other hand, the response speed of DePEGylation, in most cases, is not high enough. For example, the processes of the most studied benzoic imine bond and TACMAA-based DePEGylation need hours to complete, which may reduce the cellular uptake efficiency of the DDSs in the tumor. Additionally, the TME of each patient, or even tumor tissue, is different. As a result, for different cases, different methods of DePEGylation, cancer active targeting, and intercellular drug release should be precisely selected for precision medicine, which requires more systematic scientific and clinical research.

## Data Availability

Not applicable.

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
