# Peer review of "Drug Delivery Systems with a “Tumor-Triggered” Targeting or Intracellular Drug Release Property Based on DePEGylation"

_materials, 2022, doi:10.3390/ma15155290_

Round 1

Reviewer 1 Report

The manuscript of Ren et al. reports different approaches used for removing the “protection” due to grafted PEG molecules, called DePEGylation, from multifunctional drug delivery carriers. Different strategies and smart systems are well described in the review.

The paper is well written and supported by a fine analysis of the literature. Anyway, for considering this manuscript for publication in Materials, some issues need to be addressed:

- Since DePEGylation is not a common strategy, my suggestion is to modify the manuscript title, emphasizing more the concept of the depegylation and removing the mention of tumor/intracellular release. This could help to reach a broader audience of readers.

- Few reported studies involved an in vivo application of the obtained DDSs. A suggestion is to highlight those studies, even with a summary table at the end of the manuscript, to be recalled in the Conclusions.

- According to the impressive experience of the authors in the field, I suggest extending the section “Summary and Perspectives”, with a more in-depth analysis of the general state of the art, of the future of this strategy and analyzing more deeply the mentioned limitations.

Author Response

Dear Prof. Derniza Cozorici and the reviewers,

Many thanks for your email regarding the comments on our paper submitted to Materials.

Title: Drug Delivery Systems with a “Tumor-Triggered” Targeting or Intracellular Drug Release Property Based on DePEGylation

Manuscript ID: materials-1808006

Authors: C. Li et al.

This manuscript was revised thoroughly according to the comments and minor revisions were made as listed below. The changes are shown in red color in the revised manuscript.

Please do not hesitate to contact me should you have any questions. Thank you very much and looking forward to your kind consideration.

Best wishes,

Cao Li, Prof. PhD

Ministry-of-Education Key Laboratory for the Green Preparation and Application of Functional Materials. Hubei Key Laboratory of Polymer Materials. Hubei University

Wuhan, Hubei 430062, P. R. China

Email: licao0415@163.com

Reply to the comments:

The manuscript of Ren et al. reports different approaches used for removing the “protection” due to grafted PEG molecules, called DePEGylation, from multifunctional drug delivery carriers. Different strategies and smart systems are well described in the review.

Reply: Thanks very much.

The paper is well written and supported by a fine analysis of the literature. Anyway, for considering this manuscript for publication in Materials, some issues need to be addressed:

- Since DePEGylation is not a common strategy, my suggestion is to modify the manuscript title, emphasizing more the concept of the depegylation and removing the mention of tumor/intracellular release. This could help to reach a broader audience of readers.

Reply: Thanks for the suggestion. In fact, besides “tumor-triggered” targeting or intracellular drug release, the DePEGylation strategy has also been used in other areas in the drug delivery fields [Nanoscale Horiz. 2019, 4, 378-387]. And this review is focused on its application in dynamic protection fields. As a result, the title of the review is “Drug Delivery Systems with a “Tumor-Triggered” Targeting or Intracellular Drug Release Property Based on DePEGylation”. However, we agree with the reviewer and thanks for the suggestion. We plan to follow the suggestion to prepare a new review on the application of DePEGylation in drug delivery fields shortly.

- Few reported studies involved an in vivo application of the obtained DDSs. A suggestion is to highlight those studies, even with a summary table at the end of the manuscript, to be recalled in the Conclusions.

Reply: Thanks very much. As suggested, we offer a summary table at the end of the revised manuscript and the type of biology experiments (in vitro or in vivo) has been added to the table.

- According to the impressive experience of the authors in the field, I suggest extending the section “Summary and Perspectives”, with a more in-depth analysis of the general state of the art, of the future of this strategy and analyzing more deeply the mentioned limitations.

Reply: Thanks for the suggestion. We have prolonged the “Summary and Perspectives” section. A more in-depth analysis of the general state of the art has been added.

Reviewer 2 Report

The subject in question is definitely worth treating in a review and it will be very interesting for people working in the field of drug delivery systems, particularly with respect to cancer treatment because it describes systems that are designed to deliver drugs into tumours (and thus divert them from other tissues and reduce toxicity) and then be modified in the tumour environment so they are more likely to be taken up by tumour cells. It is interesting to have a specific review of this quite specialized topic.

 The article is well written but the English could be polished a bit.

 It has an extensive bibliography that is relevant to the subject, up to date and not containing excessive self-citation.

 It is not really relevant to say whether the conclusions are sound because the article does not describe original experimental work. The final paragraph gives a balanced summary of the state of the art and points out some areas that need further improvement.

This is an interesting review article. It could be improved by adding a summary table of the different strategies, indicating which ones have given proof of concept in vivo and which have only been tested in vitro.

It would also be helpful to have a list of abbreviations, although each one is defined at its first appearance

Author Response

Dear Prof. Derniza Cozorici and the reviewers,

Many thanks for your email regarding the comments on our paper submitted to Materials.

Title: Drug Delivery Systems with a “Tumor-Triggered” Targeting or Intracellular Drug Release Property Based on DePEGylation

Manuscript ID: materials-1808006

Authors: C. Li et al.

This manuscript was revised thoroughly according to the comments and minor revisions were made as listed below. The changes are shown in red color in the revised manuscript.

Please do not hesitate to contact me should you have any questions. Thank you very much and looking forward to your kind consideration.

Best wishes,

Cao Li, Prof. PhD

Ministry-of-Education Key Laboratory for the Green Preparation and Application of Functional Materials. Hubei Key Laboratory of Polymer Materials. Hubei University

Wuhan, Hubei 430062, P. R. China

Email: licao0415@163.com

Reply to the comments:

The subject in question is definitely worth treating in a review and it will be very interesting for people working in the field of drug delivery systems, particularly with respect to cancer treatment because it describes systems that are designed to deliver drugs into tumours (and thus divert them from other tissues and reduce toxicity) and then be modified in the tumour environment so they are more likely to be taken up by tumour cells. It is interesting to have a specific review of this quite specialized topic.

Reply: Thanks very much.

 The article is well written but the English could be polished a bit.

Reply: Thanks for the suggestion. We have tried our best to improve our English.

 It has an extensive bibliography that is relevant to the subject, up to date and not containing excessive self-citation.

Reply: Thanks very much.

 It is not really relevant to say whether the conclusions are sound because the article does not describe original experimental work. The final paragraph gives a balanced summary of the state of the art and points out some areas that need further improvement.

Reply: Thanks for the suggestion. We have prolonged the “Summary and Perspectives” section. A more in-depth analysis of the general state of the art has been added.

This is an interesting review article. It could be improved by adding a summary table of the different strategies, indicating which ones have given proof of concept in vivo and which have only been tested in vitro.

Reply: Thanks very much. As suggested, we offer a summary table at the end of the revised manuscript.

It would also be helpful to have a list of abbreviations, although each one is defined at its first appearance

Reply: Thanks very much. As suggested, a list of abbreviations has been added to the revised manuscript

Reviewer 3 Report

The authors reviewed drug delivery systems using PEG systems.  The review can be accepted after revision considering the following points:- 

  1. An in-depth discussion should be added. 
  2. The authors should create some figures to summarize some points. 
  3. The advantages and disadvantages of PEG-based systems should be included. 
  4. Literature should be summarized into Tables for the readers. 
  5. A comparison with other biopolymers should be included in these References; https://doi.org/10.3390/ijms23105405; https://doi.org/10.3390/polym13030477; https://doi.org/10.3390/polym13132086;  
  6. A prospective should be added. 
  7. The language should be improved, and typos should be corrected.

Author Response

Dear Prof. Derniza Cozorici and the reviewers,

Many thanks for your email regarding the comments on our paper submitted to Materials.

Title: Drug Delivery Systems with a “Tumor-Triggered” Targeting or Intracellular Drug Release Property Based on DePEGylation

Manuscript ID: materials-1808006

Authors: C. Li et al.

This manuscript was revised thoroughly according to the comments and minor revisions were made as listed below. The changes are shown in red color in the revised manuscript.

Please do not hesitate to contact me should you have any questions. Thank you very much and looking forward to your kind consideration.

Best wishes,

Cao Li, Prof. PhD

Ministry-of-Education Key Laboratory for the Green Preparation and Application of Functional Materials. Hubei Key Laboratory of Polymer Materials. Hubei University

Wuhan, Hubei 430062, P. R. China

Email: licao0415@163.com

Reply to the comments:

The authors reviewed drug delivery systems using PEG systems.  The review can be accepted after revision considering the following points:-

Reply: Thanks very much.

An in-depth discussion should be added.

Reply: Thanks for the suggestion. We have prolonged the “Summary and Perspectives” section. A more in-depth analysis of the general state of the art has been added.

The authors should create some figures to summarize some points.

Reply: Thanks for the suggestion. As suggested, a schematic illustration of DePEGylation for “tumor-triggered” targeting or intracellular drug release has been added to the revised manuscript.

The advantages and disadvantages of PEG-based systems should be included.

Reply: Thanks for the suggestion. We have prolonged the “Summary and Perspectives” section. The advantages and disadvantages of PEG-based systems have been added to this section in the revised manuscript.

Literature should be summarized into Tables for the readers.

Reply: Thanks very much. As suggested, we offer a summary table at the end of the revised manuscript.

A comparison with other biopolymers should be included in these References; https://doi.org/10.3390/ijms23105405; https://doi.org/10.3390/polym13030477; https://doi.org/10.3390/polym13132086; 

Reply: Thanks for the suggestion. The comparison with other biopolymers introduced in these references has been added to the “Summary and Perspectives” section in the revised manuscript.

A prospective should be added.

Reply: Thanks for the suggestion. A perspective of this art has been added to the “Summary and Perspectives” section in the revised manuscript.

The language should be improved, and typos should be corrected.

Reply: Thanks for the suggestion. We have tried our best to improve our English.

Round 2

Reviewer 3 Report

The authors addressed most of the comments and the revised version can be accepted.

Author Response

Thanks very much!